# Potential of Cucurbitacin B and Epigallocatechin Gallate as Biopesticides against *Aphis gossypii*

**DOI:** 10.3390/insects12010032

**Published:** 2021-01-05

**Authors:** Chenchen Zhao, Chao Ma, Junyu Luo, Lin Niu, Hongxia Hua, Shuai Zhang, Jinjie Cui

**Affiliations:** 1State Key Laboratory of Cotton Biology, Institute of Cotton Research, Chinese Academy of Agricultural Sciences, Anyang 455000, China; zhaochen@webmail.hzau.edu.cn (C.Z.); machaoyzu@163.com (C.M.); luojunyu1818@126.com (J.L.); nl19882006@webmail.hzau.edu.cn (L.N.); 2College of Horticulture and Plant Protection, Yangzhou University, Yangzhou 225009, China; 3Hubei Insect Resources Utilization and Sustainable Pest Management Key Laboratory, College of Plant Science and Technology, Huazhong Agricultural University, Wuhan 430072, China; huahongxia@mail.hzau.edu.cn; 4College of Plant Protection, Henan Agricultural University, Zhengzhou 450001, China

**Keywords:** plant-derived pesticide, toxicity, population-level fitness, nonhost adaptation, detoxification enzymes

## Abstract

**Simple Summary:**

The *Aphis gossypii* is a global problem for its pesticide resistance with substantial economic and ecological cost and a wide host range, including cotton and cucurbits. The development of insecticide resistance is rapid and widespread and threatens crop productivity. Biopesticides have emerged as a better alternative for pest control. Cucurbitacin B (CucB) and epigallocatechin gallate (EGCG) are the major secondary metabolites of host plants cucurbits and cotton. In this study, we used cotton- and cucurbit-specialized aphids (CO and CU) as a study system to better understand the effects of CucB and EGCG on cotton aphid. Our study showed that CucB and EGCG can significantly reduce the population-level fitness of *A. gossypii*, affect their ability to adapt to nonhost plants and alter the levels of some detoxifying enzymes, which showed a potential to be developed into new biopesticides against the notorious aphids.

**Abstract:**

*Aphis gossypii* (Glover) is distributed worldwide and causes substantial economic and ecological problems owing to its rapid reproduction and high pesticide resistance. Plant-derived cucurbitacin B (CucB) and epigallocatechin gallate (EGCG) are known to have insecticidal and repellent activities. However, their insecticidal activity on cotton- and cucurbit-specialized aphids (CO and CU), the two important host biotypes of *A. gossypii*, remains to be investigated. In the present study, we characterized, for the first time, the effects of these two plant extracts on the two host biotypes of *A. gossypii.* CucB and EGCG significantly reduced the *A. gossypii* population-level fitness and affected their ability to adapt to nonhost plants. Activities of important detoxification enzymes were also altered, indicating that pesticide resistance is weakened in the tested aphids. Our results suggest that CucB and EGCG have unique properties and may be developed as potential biopesticides for aphid control in agriculture.

## 1. Introduction

Agricultural losses due to pests pose considerable economic and ecological challenges, exacerbated by climate change [1]. So far, insect pest control has mostly depended on the use of synthetic chemical insecticides [2,3]. However, extensive and long-term application of synthetic insecticides has resulted in residual pollution in food, water, and other environmental components with adverse effects on human health and ecosystems [4], along with a strong negative impact on biodiversity [5]. Meanwhile, insects have evolved multiple mechanisms to overcome insecticide toxicity, including detoxification and excretion, which reduce pesticide effectiveness [6,7]. The development of insecticide resistance is rapid and widespread and threatens crop productivity. In contrast, biopesticides (e.g., fungus and plant extracts) have therefore emerged as a better alternative for pest control [8,9,10]. They can potentially reduce the use of chemical pesticides and provide new ideas for the synthesis of novel biopesticides for pest control [8,9,11]. Some plant extracts are being used for the management of insect pests such as *Plutella xylostella*, and *Drosophila suzukii*, and their application is becoming an integral part of ecological conservation programs [12,13].

Cucurbitacin B (CucB) and epigallocatechin gallate (EGCG) are the major secondary metabolites of host plants cucurbits and cotton (*Gossypium hirsutum* L.), respectively [14,15]. They can be easily extracted from plants or synthesized in vitro [16,17]. Cucurbitacin is a group of natural triterpenoids, oxygen-rich compounds commonly found in Cucurbitaceae family, and are toxic to some arthropods [18,19,20]. A previous study showed that high concentrations of CucB as part of an artificial diet increases aphid mortality and decreases the fitness of melon aphids [21]. EGCG is a type of catechins and is a major component of the polyphenols found in cotton tissues [14,22,23]. EGCG strongly and directly inhibits telomerase [24] and affects metabolism in mammals and insects [25,26]. Cytochrome P450 monooxygenases (P450s), glutathione S-transferases (GSTs), carboxylesterase (CarE), acetylcholinesterase (AchE), and acid phosphatase (ACP) are involved in xenobiotic metabolism of insects [27,28,29,30,31]. However, the effects of CucB and EGCG on population-level fitness and xenobiotic metabolism enzymes of cotton-melon aphids have never been adequately characterized.

The cotton-melon aphid, *Aphis gossypii* Glover (Hemiptera: Aphididae), is a notorious pest of economically important crops, including cotton and cucurbits [2,10]. It has a worldwide distribution and causes substantial crop loss owing to its rapid clonal reproduction and a broad host range [7,32,33]. It also acts as a vector for transmitting viral diseases [33,34]. Aphids live by sucking nutrients from the plant phloem, which results in nutrient loss, hinders photosynthetic capacity of plant, distorts plant growth and development, and induces rapid plant defense responses including remarkable changes in plant secondary metabolites [35]. Previous studies illustrate that *A. gossypii* populations have formed obvious host races, which use only a subset of host plant species in their recorded host range [36,37]. Cotton and cucurbit-specialized aphids (CO and CU, respectively), are two typical biotypes that have strong host-specific natures [38]. Interestingly, the two host-specialized aphids cannot survive and establish populations after reciprocal host transfers [36]. In the last few decades, *A. gossypii* has developed resistance to all major groups of synthetic insecticides and has affected cotton yields in virtually all cotton-growing areas worldwide [39]. Apparently, *A. gossypii* has evolved to adapt to sudden environmental changes, in particular, it has experienced several geographic migrations and host transfers, which is generally accompanied by multiple biotic and abiotic stresses viz. environmental stress, host nutrition stress, and population density stress. [40]. Due to the lack of active selection, cotton aphid often landed on a non-optimal host, resulting in development of strong host adaptation towards an unexpected host. Overcoming host’s secondary defense substances is critical for adaptation to an unexpected host [3,37]. Therefore, an effective control method for *A. gossypii* and understanding the mechanisms of adaptation towards an unexpected host are both essential for a potentially suitable control program. Development of biopesticides against *A. gossypii* is critical and will require a better understanding of the effects of plant secondary metabolites on aphid population dynamics.

We hypothesize that both CucB and EGCG are detrimental to the ecological fitness of cotton-melon aphid. In this study, we assess the potential effects of CucB and EGCG on population-level fitness, nonhost adaption of offspring (offspring of CU fed on cotton, and CO fed on cucumber), and activities of key detoxification enzymes of *A. gossypii* in the laboratory. The findings from the present study are essential for understanding how plant–aphid interactions are affected by these important allelochemicals and further the application of CucB or EGCG for integrated aphid management.

## 2. Materials and Methods

### 2.1. Insect Maintenance

Aphid colonies were collected from cotton and cucumber (*Cucumis sativus* L.) plants in the field of Anyang Experimental Station, Institute of Cotton Research, Chinese Academy of Agricultural Sciences (ICR, CAAS), Henan Province (36°5′34.8″ N, 114°31′47.19″ E). The colonies of cotton-specialized aphids (CO) and cucurbit-specialized aphids (CU) were maintained on cotton and cucumber plants, respectively, for more than 100 generations (2 years) without exposure to any insecticides in the laboratory. The biotypes were identified using a previously published protocol [41]. Aphids were reared in insect rearing cages (35 cm × 35 cm × 35 cm, Baoyuan Xingye Technology Co. Ltd., Beijing, China; the cage frames were made from stainless steel tube covered with nylon yarn around it) in controlled climate chambers maintained at 26 ± 1 °C with 65 ± 5% relative humidity (RH) and a photoperiod of 16:8 h (light: dark).

### 2.2. Preparation of CucB and EGCG

CucB (CAS:6199-67-3) and EGCG (CAS:989-51-5) (purity > 95% in both) were purchased from Huayueyang Biotech. Co. (Beijing, China). Acetone (CAS:67-64-1) purchased from Beijing Chemical Works (Beijing, China) was used as a solvent for CucB while EGCG was dissolved in water. Solutions of 1, 5, and 10 mg/L CucB and EGCG were prepared separately for toxicity tests according to previous studies [42,43] and our preliminary experiment. While distilled water and 50% acetone was used as a negative control, respectively.

### 2.3. Toxicity Analysis of CucB and EGCG on A. gossypii

The toxicities of CucB and EGCG on *A. gossypii* were evaluated under laboratory conditions where the aphids were fed on natural diets containing CucB and EGCG using a modified leaf-dipping method [44,45]. Briefly, individual cucumber or cotton leaves (The leaf measures 8–9 cm in diameter) were dipped in test solutions for 30 s. Treated leaves were then dried in the fume hood for 60 min. Afterward, the leaves were placed with the abaxial surface facing upward on the surface of 1.8% agar in petri dishes (9 cm D × 1.5 cm H). Aphids were then transferred to the petri dish (one in each), and the leaves were replaced every 1–2 days. Aphids were considered dead when they did not react for at least 5 min, and the body turned black after being touched with a soft brush.

#### 2.3.1. Effects of CucB and EGCG on the Life History Traits of *A. gossypii*

Five-day-old adult CO and CU (generation F) were separately transferred to the petri dishes containing host leaves treated with CucB and EGCG, and the progeny produced in 24 h was taken as F_0_ generation. After 24 h, all adult aphids (F) were removed leaving behind only the newborn nymphs (F_0_) (*n* = 100, 10 aphids for one time, and repeat for 10 times). The aphid (F_0_) survival rate and fecundity under each treatment were recorded daily (i.e., the number of newborn nymphs F_1_ laid by each aphid F_0_ and the survival status of aphids F_0_ were observed and recorded daily until the eventual death of the aphids F_0_) to estimate the direct effects of CucB and EGCG on the population-level fitness longevity and fecundity of aphid populations. All newborn nymphs (F_1_) were removed from the petri dishes immediately after birth.

#### 2.3.2. Effects of CucB and EGCG on the Nonhost Adaptation of F_1_ Generations

The CO and CU neonate aphids (F_1_) described in Section 2.3.1 were transferred to untreated nonhost plants in the petri dish (*n* = 100). In other words, the F_0_ was fed on toxin treated host plant for whole lifecycle, while the F_1_ was transferred to a clean nonhost plant. Then, the survival rate of F_1_ was recorded daily [46]. Their fecundity was checked every day until the death of all individuals to estimate the indirect effects of CucB and EGCG on the aphid population-level fitness in nonhost plant adaptation.

#### 2.3.3. Effects of CucB and EGCG on Detoxifying Enzymes of *A. Gossypii*

Following starvation for 5 h, 3rd instar nymph aphids (colonies were fed on clean host leaves) were transferred to the petri dishes containing the leaves treated with CucB and EGCG (0, 1, 5, 10 mg/L, and control) as previously described. The samples were collected to identify the content of detoxifying metabolic enzymes with different interval of exposure viz. t1; after 24 h, t2; after 48 h, and t3; after 72 h. Subsequently, aphids were immediately flash-frozen in liquid nitrogen, and stored at −80 °C until use. For each treatment/exposure time, 60 aphids were raised separately for each treatment and collected in three replicates after the specified exposure time.

The enzymes were tested using the following kits according to the instructions: Acetylcholinesterase (AChE) assay kit (Cat. No. A024-1-1), Carboxylesterase (CarE) test kit (Cat. No. A133-1-1), Glutathione S-transferase (Cat. No. GST) assay kit (Cat. No. A004), Acid phosphatase (ACP) assay kit (Cat. No. A060-1) and Insect cytochrome P450 Elisa Kit (Cat. No. H303) (Nanjing Jiancheng Bioengineering Institute, Nanjing, China).

### 2.4. Statistical Analyses

Life table data were analyzed according to the life-table, two-sex life table theory [47]. TWO-SEX-MSChart analysis was carried out for analysing the life table. The population age-specific survival rate (*l_x_*), the net reproductive rate (*R*_0_, offspring/individual), intrinsic rate of increase (*r*, day^−1^), the finite rate of increase (λ, day^−1^), and mean generation time (*T*, day) were calculated [47]. To obtain stable estimates of variances and standard errors of the developmental time, longevity, fecundity, and other population parameters, we used the bootstrap technique to calculate the means, to estimate the variances and standard error [48,49] with 100,000 resampling (100,000 bootstraps generated a normal frequency distribution and less variable results, which were not caused by the variation of sample sizes) [50]. The paired test was used to assess differences among treatments [51].

Differences in enzyme content were analyzed via one-way analysis of variance (ANOVA) followed by Tukey’s honestly significant difference (HSD) test and two-way ANOVA using SAS 9.4.

## 3. Results

### 3.1. Effects of CucB and EGCG on the Life History Traits of A. gossypii

The negative effects of CucB and EGCG on insects have been reported in *Bactrocera cucurbitae* [52], *Diabrotica virgifera* [53], *Heterodera glycines* [54], and *Drosophila*
*melanogaster* [55]. The insecticidal and repellent activities of CucB and EGCG have been studied, and an artificial diet was used to measure the effect of CucB on aphid demographics [56]. In this study, we pretreated cotton and cucumber leaves (leaf-dipping method) for aphid feeding with different concentrations of CucB and EGCG, and investigated the survival rates of the F_0_ generation of CO and CU. We found that exposure of *A. gossypii* to CucB and EGCG significantly decreased the survival rates of both biotypes (Figure 1). From day 5 of feeding, the survival rates declined rapidly in both CU and CO treated with CucB or EGCG, but declined slightly at day 8 of feeding in the water- or solvent-treatment controls. The F_0_ aphids maintained on CucB or EGCG treated leaves all died after 20–24 days of feeding, but the survival rate for CO and CU control groups were 21% and 14%, respectively. Thus, the results indicate significant toxicity for CucB and EGCG on aphids when provided as part of their natural diet. Moreover, the responses were stronger with higher concentrations of CucB and EGCG (Figure 1).

We compared the life tables of aphids treated with different concentrations of CucB and EGCG with those in control groups (Table 1 and Table 2). The parameters for population dynamics, including *r*, *R*_0_, *λ*, *T*, and oviposition day were significantly reduced when compared with the control groups except for the *T* of CO fed on EGCG treated cotton leaves (Table 2). While EGCG increased the total pre-ovipositional period (TPRP, day) (>4.2 d in CU and >5.0 d in CO), 5 mg/L CucB significantly suppressed the fecundity (22.6) and longevity (9.9 d) in CU. Furthermore, after exposing individuals to CucB and EGCG, the growth period of pre-adult and the APRP (pre-reproductive period) significantly increased. EGCG inhibited the development of F_0_ CO with more potent effects at higher concentrations (Table 1 and Table 2). However, there was no evident dose-dependent effect of EGCG on CU (Table 1 and Table 2).

### 3.2. Effects of CucB and EGCG on the Nonhost Adaptation of F_1_ Generations

Exposure to CucB significantly decreased the survival rate of *A. gossypii* offspring that fed on nonhost plants (Figure 2a,b). On cotton leaves, the survival rate of CU offspring from parents exposed to CucB was considerably decreased when compared with the control. Four days after they were transferred to nonhost plants, only 2% of CU survived while 50% survived in the control group. The results were similar for CO aphid offspring. Moreover, the effect of CucB on F_1_ was more substantial at higher concentrations. The mortality in CO aphid offspring born of the EGCG exposed parents was also increased. The survivorship curve indicated that the increased mortality occurred at early life stages (<5 d) (Figure 2d). On the contrary, EGCG improved the performance of the F_1_ of CU on non-host plants where the mortality rate of *A. gossypii* was reduced when compared to the control group (Figure 2c). Meanwhile, the mortality rate of *A. gossypii* did not increase with the concentration of EGCG. A low concentration of EGCG (1 mg/L) showed a non-host plant adaptation of cotton-melon aphid (Figure 2c,d).

We also investigated the life table parameters of the above aphid offspring in nonhost plants (Table 3 and Table 4). Compared with the untreated control group, the pre-adult duration, oviposition day, and fecundity of the F_1_ generation significantly increased, due to the treatment of the parental generation with CucB and EGCG, except F_1_ of CU on cotton after exposed to CucB (Table 3). Furthermore, after exposing the F_0_ individuals to CucB, the TPOP, longevity of the F_1_ generation of CO significantly decreased (Table 3).

Parameters for CU with maternal generation exposed to CucB (*R*_0_ less than 1, *r* < 0, and *λ* < 1) suggested a sub sequential population extinction. This effect was further enhanced when a higher concentration of CucB was used. However, compared with the control group, parental exposure to EGCG improved the fitness of CU, and this improvement was dose-dependent (*r* > 0). The values of life table parameters (*r*, *λ*) of CO did not differ significantly between CucB and EGCG treatments. Nevertheless, CucB reduced the fecundity, *R*_0_, longevity, and *T* with stronger effects at higher concentrations (Table 4). Similarly, maternal generation exposure to EGCG increased fecundity, longevity, and *T* (Table 3 and Table 4). However, 1 mg/L EGCG had stronger effects on increasing longevity, fecundity, and population parameters (*R*_0_, *λ*, *r*) in both CU and CO.

Nevertheless, when compared with the host-fed *A. gossypii*, feeding on nonhost plants significantly reduced the population-level fitness of CU and CO (Table 1 and Table 2, control). However, EGCG could improve population-level fitness of cotton-melon aphids on non-host plants to some extent (Table 3 and Table 4).

### 3.3. Effects of CucB and EGCG on Detoxifying Enzymes of A. Gossypii

#### 3.3.1. Acetylcholinesterase

The AChE activities in both biotypes of aphids fed on control leaves decreased at 48 h post-feeding and then increased slightly at 72 h, with the highest activity at 24 h (Table 5; Figure 3 and Figure 4). CU exhibited no significant differences between the control and treatment groups at 1 and 5 mg/L of CucB (Figure 3a). However, at 10 mg/L CucB treatment enhanced AChE activity (*p* < 0.05) with time. For CO, CucB lightly suppressed AChE activity at 24 h (*F*_4,10_ = 3.46, *p* = 0.051) post-feeding, but the effect was not significant at later stages Figure 3b). CU exhibited significantly increased AChE activity at 24 h and 48 h post-feeding on leaves treated with different concentrations of EGCG, but the activity returned to the level of the control group at 72 h (Figure 4a). AChE activity in CO were significantly inhibited at all three concentrations of EGCG at every time interval except for 48 h with 1 mg/L EGCG (Figure 4b). Thus, CucB and EGCG stimulated AChE activity in CU while it suppressed it in CO.

Each bar represents mean ± SE of three biological samples from different treatments. Different uppercase letters indicate significant differences at the same concentration with different treatment times, and different lowercase letters indicate significant differences at different concentrations with the same treatment times (One-way ANOVA).

#### 3.3.2. Carboxylesterase

CucB significantly decreased CarE activity in both CU and CO at all time points (Table 5; Figure 3c,d), except in CU treated with 10 mg/L CucB 24 h post-feeding (*F*_2,6_ = 0.33, *p* = 0.729) (Figure 3c). Within the cotton-melon aphids, EGCG induction significantly inhibited levels of the CarE (Table 3; Figure 4c,d). For example, CarE activities in CO were 0.03, 0.02, and 0.01 U/mg at 24 h, showing respectively 82.80%, 87.41%, and 92.50%, reductions of activity compared with the control group. Moreover, duration of EGCG exposure also significantly altered CarE levels in cotton-melon aphids (Table 5).

#### 3.3.3. Acid Phosphatase

No significant differences in ACP activity (*p* > 0.05) were found among CU fed on the plants treated varying concentrations of CucB and EGCG (Table 5; Figure 3e,f and Figure 4e). However, CucB suppressed ACP activity in CO at 48 and 72 h post-feeding. The suppression was more robust at higher concentrations (Table 5; Figure 4f).

#### 3.3.4. Cytochrome P450 Monooxygenases

The P450 activities increased significantly in CU fed on leaves treated with three different concentrations of CucB (Table 5; Figure 3g), and in CO at 24 h post-feeding fed with 10 mg/L of CucB (Figure 4h). However, the P450 activity was similar to that in the control group for CO at other time points (Table 3; Figure 3h). The treatment of EGCG induced a concentration-independent increase of P450 activity in both CU and CO when compared with the controls (Table 3; Figure 4g,h). whereas, duration of CucB and EGCG exposure showed no significantly altered P450 levels in cotton-melon aphids (Table 5).

#### 3.3.5. Glutathione S-Transferases

After *A. gossypii* was fed on leaves treated with CucB, GST activity in CO was significantly increased with increasing concentrations of CucB (Table 3; Figure 3j). However, a high concentration of CucB suppressed GST activity in CU after feeding for 72 h (Table 3; Figure 3i). Like ACP, the activity of GST in CU showed no difference at any point, except under treatment with 10 mg/L EGCG for 72 h, which was much higher than that at 24 and 48 h (Table 3; Figure 4i). A statistically significant increase in GST activity was found in the treated CO for 24 and 48 h (*p* < 0.01). With the exception of the 10 mg/L EGCG treatment, the GST levels in the treated groups decreased to the level of the control group at 72 h (Table 3; Figure 4j).

It is worth noting that after treatment with three concentrations of EGCG and two lower concentrations of CucB (1 and 5 mg/L), there was an initial increase in the overall activity of GST in CO, followed by a decrease to control group levels. When feeding on nonhost plants, the expression of detoxification-related enzymes, such as P450 and GSTs, may be activated in response to the presence of toxic substances.

## 4. Discussion

Biopesticides (natural products) have emerged as a better alternative for pest control [31]. CucB and EGCG are the major secondary metabolites of host plants cucurbits and cotton, which are the host plant of two typical biotypes of the notorious agricultural pest aphid—CU and CO [14,15,38]. Accordingly, these two natural products could provide an alternative to synthetic insecticides for environmentally-friendly control of cotton-melon aphids in future. However, the effects of EGCG and CuCB on cotton-melon aphids, especially transgenerational effect, have rarely been manipulated experimentally.

This study showed the direct effects, transgenerational effects of CucB and EGCG on the biological traits and five enzymes activity of the cotton-melon aphid. Our results clearly demonstrate that 5 mg/L CucB and 10 mg/L EGCG were effective in inhibiting development, survival, and fecundity in CU and CO. The results, thus, indicate that CucB and EGCG can directly and significantly reduce the population-level fitness of *A. gossypii* with stronger effects at higher concentrations. These results are consistent with previously published work on the toxicity of CucB [21], in which CucB was part of an artificial diet. In our study, we used a leaf-dipping method. In another study, a sub-lethal concentration of cucurbitacin E altered the percentages of survival, pupation, fecundity, and egg hatchability of *Spodoptera litura* [57]. Therefore, our results further confirm the positive effects of these plant-derived chemicals on *A. gossypii* and demonstrate their importance in defense against herbivory insect [58].

Furthermore, previous reports have suggested that cotton and cucumber serve as alternate hosts for CU and CO, respectively, under food deficiency [59]. The ability of aphids, especially host-specialized populations, to recognize and respond to nonhosts is vital for their survival and fitness [36]. Overcoming host’s secondary defense metabolites is critical for adaptation to an unexpected host [3,37]. Our results showed that cotton is not a suitable host for CU after parental exposure to CucB. CucB decreased the fitness in CO when fed with cucumber leaves with a higher concentration of CucB. However, EGCG could slightly improve the adaptability of *A. gossypii* on nonhosts. In this study, exposure to EGCG delayed the development of F_1_ of cotton-melon aphid on non-host plant, but also showed to increase its fecundity. However, it was hard to establish CU and CO populations on nonhost plants after treatment with EGCG or CucB. These results provide evidence that CucB and EGCG can indirectly affect the population-level fitness of these two host-specialized aphids following reciprocal host transfers. In practice, we needed to consider that EGCG in the field for host plants may lead to an ecological problem—the population-level fitness of non-host adaption maybe improved when exposed to EGCG.

CucB and EGCG has been recognized as an excellent organic pesticide, however, few studies have been conducted on the role of CucB and EGCG in animal subjects [21,25]. P450s, GSTs, CarE, AchE, and ACP have been reported to be involved in the xenobiotic metabolism of insects. The inhibition of AChE produces a generalized synaptic collapse that can lead to insect death [30]. AChE is responsible for catalyzing the hydrolysis of the neurotransmitter acetylcholine leading to the release of acetate and choline [60,61]. ACP catalyzes the hydrolysis of phosphoric acid to produce inorganic phosphoric acid under acidic conditions, which is often related to nutrient uptake [62]. P450 enzymes have important roles in the synthesis and degradation of ecdysteroids and juvenile hormones and in the metabolism of foreign chemicals of natural or synthetic origin [63]. GSTs belong to a multigene family of dimeric, multifunctional proteins that have a central role in detoxification of xenobiotic compounds, including drugs, herbicides, and insecticides [64]. Furthermore, the elevated levels of GSTs are associated with tolerance to insecticides [65].

It is worth noting that after treatment with three concentrations of EGCG and two lower concentrations of CucB (1 and 5 mg/L), there was an initial increase in the overall activity of GST in CO, followed by a decrease to the control group levels. When feeding on nonhost plants, the expression of detoxification-related enzymes, such as P450 and GSTs, may be activated in response to the presence of toxic substances [66]. We speculate that feeding on a natural diet containing CucB and EGCG causes changes in the expression of insect detoxification enzymes or enzyme activities as well as related detoxification mechanisms, which would further weaken the ability of aphids to metabolize. In plants, P450 enzymes involved in secondary metabolism plays a role in cucurbitacin biosynthesis [67,68]. CucB and EGCG significantly altered the levels of few detoxifying enzymes. They increased the levels of AChE and P450 in CU, decreased the levels of AChE (in CO), CarE, and ACP. EGCG and CucB were found to increase activity of P450 during the early feeding stage (24 h), but diminished the activity to normal levels at 48 and 72 h. The alteration in GST activity varied with the concentration of CucB. GST activity was stimulated by CucB in CO, while a contrasting effect was observed in CU with a gradual increase in feeding time. Low concentrations of EGCG increased GST activity up until 48 h in CO, but did not affect GST activity in CU. Interestingly, most detoxification enzymes in cotton-melon can be significantly altered by low concentrations of EGCG and CucB within short duration, which showed potential in preventing and controlling this pest. Our results showed an overwhelming inhibition of CarE activity by CucB and EGCG, indicating loss of detoxifying ability and increased vulnerability to stressors in CU and CO. Such an effect may be due to the absence of a mechanism to metabolize CucB and EGCG in *A. gossypii*. We speculate that feeding on a natural diet containing CucB and EGCG causes changes in the expression of insect detoxification enzymes or enzyme activities as well as related detoxification mechanisms, which would further weaken the ability of aphids to metabolize.

Hereby, we conclude that differences in biochemical responses of aphids to xenobiotics are influenced by a variety of factors such as the nature and dosage of the test substance, the age and biotype of the insect, and the type of enzyme being assayed. Our results showed that CucB and EGCG had direct and indirect impact on cotton-melon aphids.

## 5. Conclusions

This article aimed to provide a preliminary study on the toxic effects of CucB and EGCG on *A. gossypii* with the focus on population-level fitness, and detoxification enzymes to theoretical support for the subsequent development of pesticides using them as effective ingredients. In summary, CucB and EGCG can significantly reduce the population-level fitness of *A. gossypii*, affect their ability to adapt to nonhost plants and alter the levels of some detoxifying enzymes. Based on the presented data and subsequent results, CucB and EGCG have the potential to be developed into new biopesticides against the notorious aphids. A future study is necessary to exploit these potential biopesticides through a comparative evaluation with other known bio-pesticides, especially the transgenerational effects of EGCG. This may also have important implications in pest management efforts to control *A. gossypii*.

## Figures and Tables

**Figure 1 insects-12-00032-f001:**
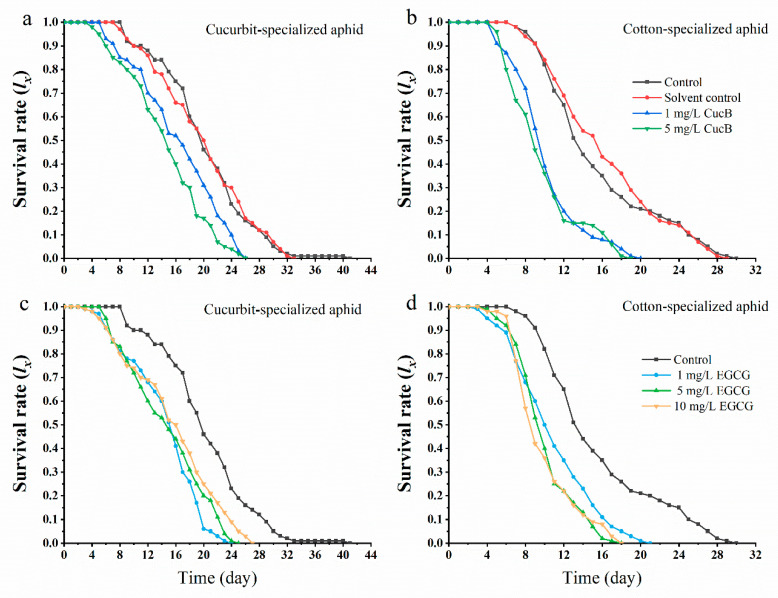
Survival rate (*l_x_*) of *Aphis gossypii* exposed to Cucurbitacin B (CucB) and epigallocatechin gallate (EGCG). (**a**): CUS exposed to CucB; (**b**): COS exposed to CucB; (**c**): CUS exposed to EGCG; (**d**): COS exposed to EGCG.

**Figure 2 insects-12-00032-f002:**
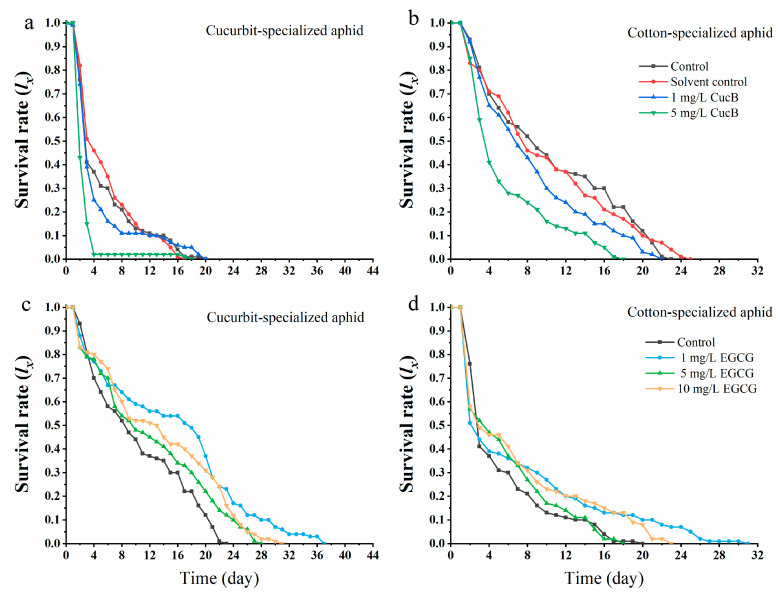
Survival rate (*l_x_*) of *Aphis gossypii* on nonhost plants when the maternal generation is exposed to CucB and EGCG. (**a**): CUS exposed to CucB; (**b**): COS exposed to CucB; (**c**): CUS exposed to EGCG; (**d**): COS exposed to EGCG.

**Figure 3 insects-12-00032-f003:**
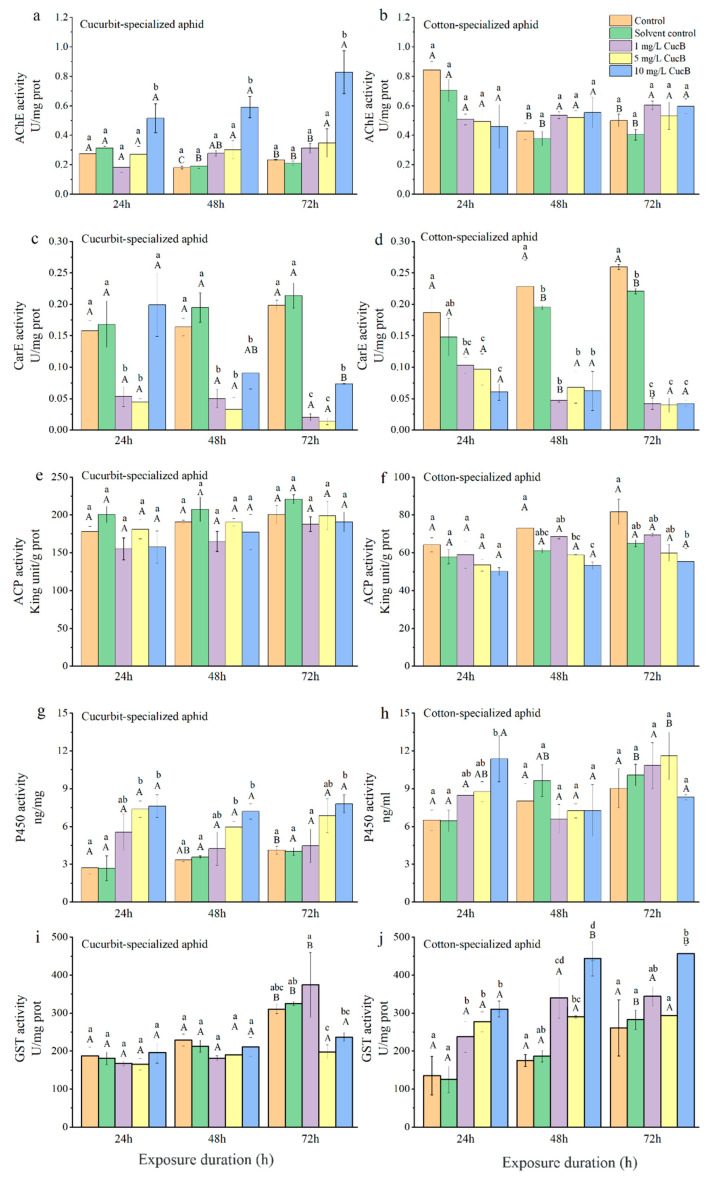
Effect of CucB on enzyme activity in *Aphis gossypii* in different time intervals. (**a**): AChE activity of CUS; (**b**): AChE activity of COS; (**c**): CarE activity of CUS; (**d**): CarE activity of COS; (**e**): ACP activity of CUS; (**f**): ACP activity of COS; (**g**): P450 activity of CUS; (**h**): P450 activity of COS; (**i**): GST activity of CUS; (**j**): GST activity of COS. Bars labelled with different lowercase letters indicate significant differences between concentrations in same time interval, bars labelled with different capital letters indicate significant differences between time intervals in in same concentration (*p* < 0.05).

**Figure 4 insects-12-00032-f004:**
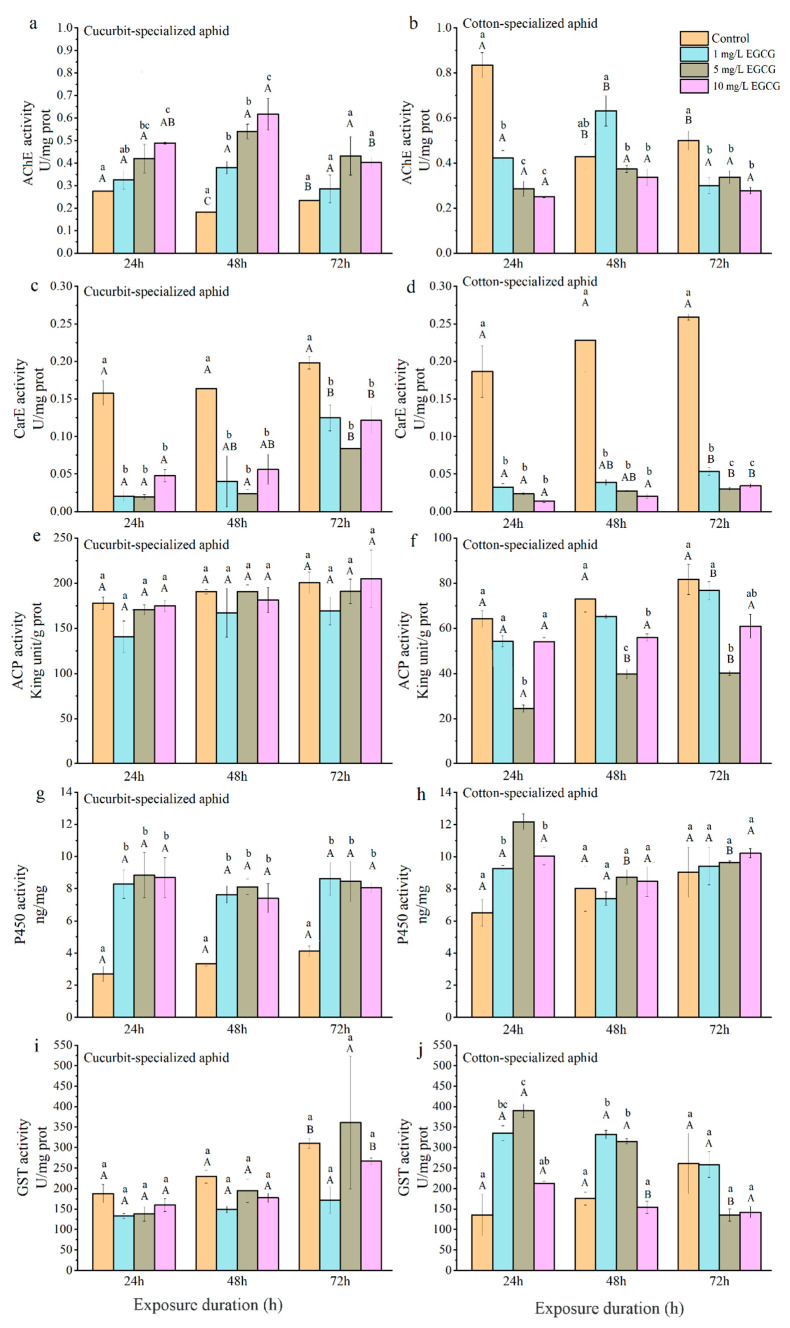
(**a**): AChE activity of CUS; (**b**): AChE activity of COS; (**c**): CarE activity of CUS; (**d**): CarE activity of COS; (**e**): ACP activity of CUS; (**f**): ACP activity of COS; (**g**): P450 activity of CUS; (**h**): P450 activity of COS; (**i**): GST activity of CUS; (**j**): GST activity of COS. Effect of EGCG on enzyme activity in *Aphis gossypii* in different time intervals.Bars labelled with different lowercase letters indicate significant differences between concentrations in same time interval, bars labelled with different capital letters indicate significant differences between time intervals in in same concentration (*p* < 0.05).

**Table 1 insects-12-00032-t001:** Effect of CucB and EGCG on fecundity, longevity, adult pre-reproductive period (APRP), total pre-reproductive period (TPRP), and mean and oviposition day of *Aphis gossypii.*

Aphid	Parameters	Cucurbitacin B (CucB)				Epigallocatechin Gallate (EGCG)			
		Control	Solvent Control	1 mg/L CucB	5 mg/L CucB	Control	1 mg/L EGCG	5 mg/L EGCG	10 mg/L EGCG
**Cucurbit-specialized** **aphids**	Pre-adult duration	4.02 ± 0.02 a	4.03 ± 0.03 a	4.16 ± 0.05 b	4.17 ± 0.0 b	4.02 ± 0.02 a	4.05 ± 0.10 ab	4.10 ± 0.03 b	4.20 ± 0.04 b
	APRP	0.67 ± 0.05 a	0.76 ± 0.05 a	0.94 ± 0.04 b	0.89 ± 0.04 b	0.67 ± 0.05 a	0.71 ± 0.04 b	0.72 ± 0.04 b	0.74 ± 0.03 b
	Fecundity	47.49 ± 1.33 a	50.05 ± 1.62 a	26.78 ± 1.80 b	22.56 ± 1.40 c	47.49 ± 1.33 a	32.64 ± 1.92 b	32.86 ± 1.52 b	30.04 ± 1.45 b
	Longevity	15.7 ± 0.59 a	16.4 ± 0.58 a	10.27 ± 0.35 c	9.9 ± 0.37 c	15.7 ± 0.59 a	11.0 ± 0.41 b	10.2 ± 0.30 b	10.0 ± 0.32 b
	TPRP	4.7 ± 0.05 a	4.8 ± 0.05 b	4.9 ± 0.04 c	4.9 ± 0.04 c	4.2 ± 0.05 a	4.2 ± 0.05 a	4.6 ± 0.04 b	4.7 ± 0.05 b
	Oviposition day	11.0 ± 0.45 a	9.9 ± 0.39 a	5.35 ± 0.25 b	4.8 ± 0.31 b	9.9 ± 0.39 a	6.6 ± 0.36 b	6.0 ± 0.28 b	5.9 ± 0.30 b
**Cotton-specialized** **aphids**	Pre-adult duration	4.80 ± 0.04 a	4.98 ± 0.01 b	5.00 ± 0.01 b	5.00 ± 0.02 b	4.8 ± 0.04 a	5.89 ± 0.03 c	5.69 ± 0.05 b	6.00 ± 0.01 d
	APRP	0.24 ± 0.05 a	0.37 ± 0.05 ab	0.34 ± 0.05 ab	0.80 ± 0.04 b	0.24 ± 0.05 a	0.71 ± 0.05 c	0.60 ± 0.07 b	1.63 ± 0.12 d
	Fecundity	37.64 ± 0.48 a	39.43 ± 0.76 a	36.80 ± 1.43 a	29.87 ± 1.12 b	37.64 ± 0.69 a	18.10 ± 0.78 b	14.42 ± 0.68 c	15.26 ± 0.88 c
	Longevity	20.4 ± 0.63 a	20.1 ± 0.66 a	16.5 ± 0.59 b	14.7 ± 0.56 c	20.4 ± 0.64 a	14.5 ± 0.50 b	14.9 ± 0.56 b	15.7 ± 0.64 b
	TPRP	5.0 ± 0.04 a	5.8 ± 0.04 b	5.3 ± 0.05 b	5.4 ± 0.05 b	5.0 ± 0.04 a	6.6 ± 0.05 c	6.3 ± 0.07 b	7.6 ± 0.12 d
	Oviposition day	13.6 ± 0.36 a	12.8 ± 0.45 b	10.4 ± 0.48 c	8.5 ± 0.36 d	13.6 ± 0.36 a	8.4 ± 0.36 b	7.6 ± 0.39 b	9.3 ± 0.41 b

Standard errors (SE) were estimated using the bootstrap technique with 100,000 re-samplings. Means followed by different lowercase letters (a, b, c, d) in the same row are significantly different between treatments, determined by using the paired bootstrap test in TWOSEX-MSChart *(p* < 0.05).

**Table 2 insects-12-00032-t002:** Effects of CucB and EGCG on population parameters of *Aphis gossypii.*

Aphid	Parameters	CucB				EGCG			
		Control	Solvent Control	1 mg/L CucB	5 mg/L CucB	Control	1 mg/L EGCG	5 mg/L EGCG	10 mg/L EGCG
**Cucurbit-specialized** **aphids**	*λ*	1.5659 ± 0.003 a	1.5371 ± 0.003 b	1.5110 ± 0.009 c	1.4802 ± 0.009 d	1.5659 ± 0.003 a	1.5049 ± 0.009 b	1.5885 ± 0.007 a	1.5702 ± 0.006 a
	*R* _0_	47.49 ± 1.33 a	50.05 ± 1.62 a	25.44 ± 1.26 c	22.33 ± 1.40 c	47.49 ± 1.33 a	31.01 ± 1.96 b	32.53 ± 1.54 b	29.44 ± 1.49 b
	*r*	0.4484 ± 0.002 a	0.4299 ± 0.002 b	0.4128 ± 0.005 c	0.3921 ± 0.006 d	0.4484 ± 0.002 a	0.4087 ± 0.006 b	0.4628 ± 0.004 a	0.4512 ± 0.004 a
	*T*	8.6 ± 0.07 b	9.1 ± 0.08 a	7.8 ± 0.08 c	7.9 ± 0.08 c	8.6 ± 0.07 a	8.4 ± 0.10 a	7.5 ± 0.07 b	7.5 ± 0.10 b
**Cotton-specialized** **aphids**	*λ*	1.5032 ± 0.004 a	1.4772 ± 0.003 b	1.5063 ± 0.006 a	1.4697 ± 0.007 b	1.5032 ± 0.004 a	1.2988 ± 0.006 b	1.2925 ± 0.005 b	1.23272 ± 0.006 c
	*R* _0_	37.64 ± 0.69 a	39.42 ± 0.87 a	36.80 ± 1.42 a	28.38 ± 1.24 b	37.64 ± 0.69 a	16.47 ± 0.87 b	13.84 ± 0.71 c	13.89 ± 0.91 c
	*r*	0.4076 ± 0.003 a	0.3901 ± 0.002 b	0.4097 ± 0.005 a	0.3851 ± 0.05 b	0.4076 ± 0.003 a	0.2614 ± 0.004 b	0.2566 ± 0.004 b	0.2092 ± 0.006 c
	*T*	8.9 ± 0.07 b	9.4 ± 0.06 a	8.8 ± 0.07 bc	8.7 ± 0.04 c	8.9 ± 0.07 a	10.7 ± 0.10 c	10.2 ± 0.11 b	12.6 ± 0.19 d

*r*: intrinsic rate of increase; λ: finite rate of increase; *R*_0_: net reproductive rate; *T*: mean generation time (offspring/individual); standard errors (SE) were estimated using the bootstrap technique with 100,000 re-samplings; differences between two treatments were compared using a paired bootstrap test implemented in TWOSEX-MSChart. The means in the same rows followed by different lowercase letters (a, b, c, d) indicate significant differences between treatments (*p* < 0.05).

**Table 3 insects-12-00032-t003:** Effect of CucB and EGCG on fecundity, longevity, adult pre-reproductive period (APRP), total pre-reproductive period (TPRP), and mean and oviposition day of *Aphis gossypii* on nonhost plant.

Treated aphid	Parameters	CucB				EGCG			
		Control	Solvent Control	1 mg/L CucB	5 mg/L CucB	Control	1 mg/L EGCG	5 mg/L EGCG	10 mg/L EGCG
**Cucurbit-specialized** **aphids transferred to cotton**	Pre-adult duration	5.00 ± 0.01 a	5.94 ± 0.06 b	6.02 ± 0.05 b	6.05 ± 0.06 b	5.00 ± 0.01 a	6.01 ± 0.01 b	6.03 ± 0.03 b	6.00 ± 0.01 b
	APRP	2.58 ± 0.43 a	2.31 ± 0.53 a	3.23 ± 0.37 a	4.00 ± 0.75 a	2.58 ± 0.43 a	1.42 ± 0.09 b	0.38 ± 0.12 c	1.25 ± 0.07 b
	TPOP	7.6 ± 0.54 a	9.2 ± 0.38 b	8.2 ± 0.56 a	10.0 ± 0.76 b	7.9 ± 0.54 a	6.4 ± 0.13 b	6.3 ± 0.08 b	6.4 ± 0.10 b
	Fecundity	0.7 ± 0.20 a	0.7 ± 0.20 a	4.6 ± 0.84 b	5.5 ± 1.13 b	0.7 ± 0.20 a	9.4 ± 0.84 c	5.8 ± 0.71 b	6.4 ± 0.64 b
	Longevity	5.5 ± 0.44 a	5.8 ± 0.41 a	4.9 ± 0.45 a	2.9 ± 0.20 b	5.5 ± 0.44 a	7.3 ± 0.76 b	6.0 ± 0.47 ab	7.1 ± 0.66 b
	Oviposition day	1.8 ± 0.22 a	1.5 ± 0.24 a	2.4 ± 0.38 b	4.0 ± 0.76 b	1.8 ± 0.22 a	6.9 ± 0.66 c	4.1 ± 0.38 b	5.1 ± 0.43 b
**Cotton- specialized aphids transferred to cucumber**	Pre-adult duration	4.76 ± 0.08 a	4.92 ± 0.79 a	4.13 ± 0.05 b	4.26 ± 0.06 b	4.92 ± 0.03 a	6.12 ± 0.04 c	6.00 ± 0.01 b	6.37 ± 0.07 d
	APRP	0.75 ± 0.11 a	0.62 ± 0.08 a	0.48 ± 0.09 a	0.50 ± 0.08 a	0.75 ± 0.11 a	1.47 ± 0.10 b	0.92 ± 0.12 a	1.33 ± 0.13 b
	TPOP	5.7 ± 0.12 a	5.4 ± 0.10 a	4.7 ± 0.09 b	4.2 ± 0.06 c	5.7 ± 0.12 a	7.5 ± 0.10 c	7.0 ± 0.12 b	7.7 ± 0.16 c
	Fecundity	12.6 ± 1.13 a	11.3 ± 0.91 ab	9.3 ± 0.77 bc	8.5 ± 0.77 c	12.6 ± 1.13 a	21.8 ± 1.24 b	13.3 ± 1.15 a	15.7 ± 1.29 a
	Longevity	10.5 ± 0.69 a	10.1 ± 0.69 ab	8.6 ± 0.57 b	6.0 ± 0.45 c	10.5 ± 0.69 a	15.3 ± 1.02 b	12.2 ± 0.83 ab	13.5 ± 0.88 b
	Oviposition day	8.5 ± 0.64 a	7.3 ± 0.55 ab	6.4 ± 0.48 bc	5.3 ± 0.5 c	8.5 ± 0.64 a	8.4 ± 0.62 a	12.6 ± 0.69 b	9.8 ± 0.67 a

Standard errors (SE) were estimated using the bootstrap technique with 100,000 re-samplings. Means followed by different lowercase letters (a, b, c, d) in the same row are significantly different between treatments, determined by using the paired bootstrap test in TWOSEX-MSChart *(p* < 0.05).

**Table 4 insects-12-00032-t004:** Effects of CucB and EGCG on F_1_ population parameters of *Aphis gossypii* on nonhost plant.

Treated aphid	Parameters	CucB				EGCG			
		Control	Solvent Control	1 mg/L CucB	5 mg/L CucB	Control	1 mg/L EGCG	5 mg/L EGCG	10 mg/L EGCG
**Cucurbit-specialized** **aphids transferred to cotton**	*λ*	0.8664 ± 0.03 a	0.8808 ± 0.003 a	0.97218 ± 0.03 b	0.8521 ± 0.03 a	0.8664 ± 0.03 a	1.1153 ± 0.02 b	1.0845 ± 0.02 b	1.0962 ± 0.02 b
	*R* _0_	0.23 ± 0.07 a	0.74 ± 0.21 b	0.24 ± 0.08 a	0.11 ± 0.07 b	0.23 ± 0.07 a	3.39 ± 0.54 b	2.16 ± 0.38 b	2.64 ± 0.41 b
	*r*	−0.1434 ± 0.04 a	−0.1269 ± 0.03 a	−0.0270 ± 0.02 b	−0.1601 ± 0.04 a	−0.1433 ± 0.04 a	0.1091 ± 0.01 b	0.0812 ± 0.02 b	0.0918 ± 0.01 b
	*T*	10.2 ± 0.63 a	11.2 ± 0.43 a	10.7 ± 1.07 b	13.8 ± 0.17 b	10.2 ± 0.63 ab	11.2 ± 0.47 a	9.5 ± 0.22 b	10.6 ± 0.29 a
**Cotton- specialized aphids transferred to cucumber**	*λ*	1.2144 ± 0.01 a	1.2207 ± 0.01 a	1.2200 ± 0.01 a	1.1708 ± 0.02 a	1.2144 ± 0.013 a	1.2103 ± 0.008 a	1.1946 ± 0.010 a	1.1966 ± 0.009 a
	*R* _0_	8.18 ± 0.95 a	7.81 ± 0.82 ab	5.74 ± 0.65 b	3.38 ± 0.51 b	8.18 ± 0.95 a	14.58 ± 1.32 c	9.07 ± 1.00 ab	11.48 ± 1.17 bc
	*r*	0.1942 ± 0.011 a	0.1994 ± 0.009 a	0.1989 ± 0.012 a	0.1577 ± 0.020 a	0.1942 ± 0.011 a	0.1908 ± 0.006 a	0.1778 ± 0.008 a	0.1795 ± 0.008 a
	*T*	10.8 ± 0.19 a	10.3 ± 0.28 a	8.8 ± 0.21 b	7.7 ± 0.19 c	10.8 ± 0.19 a	14.0 ± 0.20d	12.4 ± 0.25 b	13.6 ± 0.27 c

*r*: intrinsic rate of increase; λ: finite rate of increase; *R*_0_: net reproductive rate; *T*: mean generation time (offspring/individual); standard errors (SE) were estimated using the bootstrap technique with 100,000 re-samplings; differences between two treatments were compared using a paired bootstrap test implemented in TWOSEX-MSChart. The means in the same rows followed by different lowercase letters indicate significant differences between treatments (*p* < 0.05).

**Table 5 insects-12-00032-t005:** Effects of concentration and the duration exposure of CucB on EGCG on detoxifying enzyme activities.

Detoxifying enzyme	Aphid	CucB (*p*)			EGCG (*p*)		
		Concentration	Duration	Interaction	Concentration	Duration	Interaction
AChE	CU	**<0.0001**	0.0884	0.0861	**<0.0001**	**0.0298**	0.1113
	CO	0.5504	**0.0388**	**0.0044**	**<0.0001**	**0.0035**	**<0.0001**
CarE	CU	**<0.0001**	0.2753	**0.0133**	**<0.0001**	**<0.0001**	0.5676
	CO	**<0.0001**	0.9971	**0.0157**	**<0.0001**	**0.0442**	0.5438
ACP	CU	0.2109	**0.0227**	0.9992	0.0913	0.0881	0.9895
	CO	**<0.0001**	**0.0036**	0.8920	**<0.0001**	**<0.0001**	0.3220
P450	CU	**<0.0001**	0.5598	0.8144	**<0.0001**	0.5028	0.8965
	CO	0.7502	**00.0282**	0.0909	**0.0124**	**0.0412**	0.0890
GST	CU	0.1188	**<0.0001**	0.0871	0.1413	**0.0055**	0.6950
	CO	**<0.0001**	**0.0019**	0.2817	**<0.0001**	**0.0104**	**0.0002**

Bolded values denote significant differences at α = 0.05 (Two-way analysis of variance).

## Data Availability

The data presented in this study are available on request from the corresponding author.

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
