# Peer review of "Potential of Cucurbitacin B and Epigallocatechin Gallate as Biopesticides against Aphis gossypii"

_insects, 2021, doi:10.3390/insects12010032_

Round 1

Reviewer 1 Report

Reviewer comments

General comments: This is a well-written study that aims to inform us of the effects of secondary metabolites on the life history traits of A. gossypii. The topic is very interesting, but I think the authors need to improve their manuscript by major changes. Accordingly, the discussion part needs some improvement and some more references. Please see my specific comments below.

I think this paper is written, with a clean methodology, results, and discussion. The tables and figures are well done and intuitively presented. I have provided major comments and suggestions for the authors. And authors need to improve their manuscript accordingly.

Introduction

-Line 28-29, 36: The authors need to change the reference (2, 3, and 7). These three are not related to their statement. The authors need to add the most relevant reference in their statement.

-Line 43: change “cucurbit” to “Cucurbitaceae family”

-Line 43: change “studies” to “a previous study”

-Line 52: need to add order and family name when the authors use it for the first time. Kindly follow this for the whole manuscript. 

-Line 52-54: need to add the proper reference for these two statements.

-Line 60: In the whole manuscript authors used some references like (Liu et al., 2008). It's not an insects reference style. Authors need to follow the insects' reference style for their manuscript.

Materials and methods

-Line 76: need to add the scientific name of cotton and cucumber plants.

-Line 79: change “COS and CUS” to “CO and CU for the whole manuscript”

-Line 81: change “described” to “published protocol”

-Line 82: add ‘’the size of the cage’’

-Line 92: add ‘’the adults of ’’before A. gossypii

-Line 94: ‘’What is the drug membrane method’’?

-Line 96: add ‘’the diameter of the leaf size’’

-Line 99: How many replicates have been done for this experiment? Authors need to mention this in the manuscript.

-Line 99: What about the post-treatment time checking the mortality of A. gossypii?

-Line 101: change all the ‘’subsections’’ in ‘’italic’’ for the whole manuscript.

-Line 102: How many F generations used in this study? The authors need to mention it.

-Line 108: change “population fitness of aphids” to “longevity and fecundity of aphid populations”

-For the life history traits of F0 and F1 generation the authors used 100 aphids. Did the authors conduct the life history traits experiment with 100 aphids at the same time?

-Line 110: change “F1 nonhost adaptation of A. gossypii” to “the nonhost adaptation of F1 generations”

-Line 118: Why the authors starved the aphis populations for 5 h?

-Line 124: After this section authors need to add another subsection of protein extraction. How they extracted the protein from aphid? What about the protein concentration?

Results and discussion

I am strongly recommended the authors separate the results and discussion part into two sections. And need to improve both parts based on their result. In the discussion part, the authors need to compare their results with other relevant results.

-Line 141: Authors need to use the same section title for the result part as they used for the materials and methods.

-Where is the toxicity result of CucB and EGCG? As they mention in the 2.3 section of materials and methods.

-In this manuscript, the authors used three different concentrations of CucB and EGCG, so where are the LC50 values? Authors need to perform the probit analysis for the calculation of the LC50 values.

-In this manuscript, the authors checked the life history traits of the F0 generation, but they did not mention the developmental days of every stage of the aphid population. The authors need to add a separate table of the developmental time (days) of the F0 progeny. Authors need to follow the recently published paper by (Mostafiz, M.M.; Alam, M.B.; Chi, H.; Hassan, E.; Shim, J.-K.; Lee, K.-Y. Effects of Sublethal Doses of Methyl Benzoate on the Life History Traits and Acetylcholinesterase (AChE) Activity of Aphis gossypii. Agronomy 2020, 10, 1313.) as a reference to improve their manuscript. 

-Why the authors did not use the concentration of 10 mg/mL for CucB?

-If there is no difference between the concentrations of CucB and EGCG, I would suggest using the most significant concentration in the life history experiment.

-The authors need to make a separate table with r; λ; R0; and T only. Try to follow (Mostafiz, M.M.; Alam, M.B.; Chi, H.; Hassan, E.; Shim, J.-K.; Lee, K.-Y. Effects of Sublethal Doses of Methyl Benzoate on the Life History Traits and Acetylcholinesterase (AChE) Activity of Aphis gossypii. Agronomy 2020, 10, 1313.) as a reference to improve the manuscript. 

-In the Y-axis title of Figures 1 and 2, the authors need to add (lx) after the survival rate.

-What about the pre-adult stage and APRP (adult pre-reproductive period) of the F0 and F1 generation?

- change “TPOP” to “TPRP’’

-Authors need to add a footnote for each table.

-Need abbreviation if they used a term for the first time likely TPRP (total pre-reproductive period)

-Authors need to add an X-axis title for Figures 3 and 4.

-Why authors checked the enzyme activity after 24, 48, and 72 h later? Why not 1 h, 6 h, or 12 h? Maybe the pesticides can show rapid inhibition activity after the feeding?

-The results part of the enzyme activity is too much complex. Authors need to focus on the key point of the enzyme activity so that the reader can easily understand the differences.

-Why the units of the enzyme activity results are different from each other?

References

The reference section is not accordingly to the journal style. The authors need to pay more attention to it and make consistency for each reference.

Author Response

Dear Prof:

First of all, we sincerely say sorry to the editor and reviewers. We admit that there are a lot of shortcomings and deficiencies in our manuscript. What's worse is that my student and I were very anxious and in a hurry when writing this paper in order to apply for the chance to become a Ph.D. candidate, so that the quality of the paper is not satisfactory. Fortunately, after receiving the editor and reviewers' comments, we were deeply aware of the inadequacy of this paper, so that we tried our best, according to the comments, to optimize the contents of the figures, re-analyze and re-write the results section to convey a central message, and improve the operability of materials & methods. We also improved the depth and pertinence of the discussion section. We hope that our improvement can be felt, and we are willing to accept more comments and suggestions so that our articles can be published.

We would like to express our sincere thanks to the reviewers for the constructive and positive comments. Point by point responses to the reviewers’ comments are listed below this letter, which were marked by blue.

We hope that the revised version of the manuscript is now acceptable for publication in your journal.

I look forward to hearing from you soon. 

With best wishes, 

Yours sincerely,

Chenchen Zhao

Reviewer 2 Report

Comments to the Authors

Potential of Cucurbitacin B and Epigallocatechin gallate as biopesticides against Aphis gossypii

General Comments:

Zhao et al., tested the effect of CucB and EGCG on the cotton- and cucurbit-specialized aphids (CUS and COS), Aphis gossypii. They showed that CucB and EGCG both reduce aphid survivorships as well as fitness on their corresponding host plants, they demonstrated that CucB and EGCG also altered the level of detoxifying enzymes. In addition, they also test the effects of CucB and EGCG on aphid adaptation to non-host plants. The presentation of the paper is in general clear. However, some more details, explanations, presentation of the data need to be added or changed prior to publication. I list my comments below and I hope my comments will help to improve the manuscript

Major comments:

  1. My major concern is about the work for the effects of CucB and EGCG on non-host plants. First, it needs more introduction to explain the logic and the importance of these non-host plant experiments. Second, as shown in figure 2c, 2d, the data showed that the EGCG actually improved the performance of nymphs on non-host plants. The current presentation of data (line 194-197) is confusing or even contradictory to the results in the figure. Third, because of the different effects of EGCG and CucB on aphids for non-host plants, I think the authors needs to make conclusions for EGCG and CucB separately. To me, the use of EGCG in the field for host plants may lead to an ecological problem for non-host plants and beyond. Fourth, I am not convinced that they can conclude anything about aphid adaptation on non-host plants by data only from one generation. In addition, in line 210-211, the author state that “CUS and COS failed to establish their populations on non-hot plants after treatment with EGCG and CucB”. More data needs to be provided to support this statement. From the data presentation in Figure 2c and 2d, the aphids all survived >15 days, during which the aphids could already produce F2 and even F3 nymphs.
  2. For aphid bioassays presented both in Figure 1 and Figure 2 (method 2.3.1 and method 2.3.2), I am concerned if the experiment is only done once. Also, it is not clear how many experimental replicates were done for other experiments.

Other comments:

  1. All figures need legends; and statistical results need to be denoted on the figures for the survival rates.
  2. For the CucB results, except the Solvent control, there is another Control (black line in the figure1&2), please explain what is this control?
  3. The authors selected different concentrations on testing the effects of CucB and EGCG on aphids. Why those specific concentrations were used, please clarify and what does it mean for those concentrations (e.g. 5 mg/L) in terms of application in the field?
  4. Figure 3 and 4: the resolution is too low. The denotation of statistical results needs to be changed, the current denotation is hard to read (may be make the upper-case and lower-case letters on different levels instead of side by side?)
  5. “Population fitness”: may be “fitness at population level” or “population-level fitness”? needs to be changed throughout the manuscript.
  6. The difference in the current study to the study of citation 18 needs to be fully discussed.
  7. Line 56-61: the logic of testing enzymes such as P450s, GSTs, CarE etc needs more explanation. Why those enzymes are relevant to CucB and EGCG treatment? What about other enzymes or pathways?  
  8. Line 136: “bootstrap technique” needs to be fully explained.
  9. There are some typos etc needs to be checked throughout the manuscript: e.g:

Line 18: “effects of Effect of …”

Line 87: “Chinese Academy of Agricultural Science” to be “Chinese Academy of Agricultural Sciences”.

Line 261: “otton-melon”

Author Response

(The authors gave the same response as above.)

Round 2

Reviewer 1 Report

The manuscript described a technically sound piece of scientific research with data that supports the conclusions. The conclusion is properly written on the basis of the presented data. The statistical analysis has been performed appropriately and rigorously. The manuscript is presented in an understandable manner and written in standard English.

Author Response

Thank you very much for your recognition.

Happy holidays!

Reviewer 2 Report

Comments to the Authors R1

Potential of Cucurbitacin B and Epigallocatechin gallate as biopesticides against Aphis gossypii

General Comments:

In the revised version, the authors dealt with/explained most of my comments and concerns. However, the explanations for some of my concerns needs to go beyond just a response to reviewer, they need to be reflected in the manuscript.

Line numbers in my comments refer to document: insects-1028484-r1 (with track-changes)

Major comments:

  1. The results for F2 and F3 results need to be presented in the manuscript, maybe incorporate with the current F1 results in a comprehensive way.
  2. Line 128: “(n=100, 10 aphids for one time, and repeat for 10 times)”, the variations for the 10-time repeats needs to be added to the figures.
  3. Line 161: “bootstrap technique”, the current statement is still not clear. Please add a full explanation on how the bootstrap was done.
  4. The last section of discussion is independent from results, it is better to make a seperate “Discussion” section.

Some other minor comments:

Line 18: “(CU and CO)” to “(CO and CU)”

Line 417: “CO and CO”?

Line 440: “expose to and EGCG”, remove the “and”

Line 440: “the F1 of cotton-melon aphid” to “of the F1 cotton-melon aphid”

Author Response

Dear Professor:

Thank you very much for your letter and advice. We have revised the manuscript, and would like to re-submit it for your consideration. We have addressed the comments raised by the you, and the amendments are highlighted in red in the revised manuscript. Point by point responses to the comments are listed below this letter, which was marked in blue.

We would like to express our sincere thanks to you for the constructive and positive comments.
